# Effects of Bi_2_O_3_ Doping on the Mechanical Properties of PbO Ceramic Pellets Used in Lead-Cooled Fast Reactors

**DOI:** 10.3390/ma12121948

**Published:** 2019-06-17

**Authors:** Yan Ma, Anxia Yang, Huiping Zhu, Arslan Muhammad, Pengwei Yang, Bokun Huang, Fenglei Niu

**Affiliations:** Key Laboratory of Passive Nuclear Power Safety and Technology, North China Electric Power University, Beijing 102206, China; anxiayang@foxmail.com (A.Y.); zhuhuiping@ncepu.edu.cn (H.Z.); muhammad.arslan@hotmail.com (A.M.); 18810676955@163.com (P.Y.); ncepuhbk@163.com (B.H.); niufenglei@ncepu.edu.cn (F.N.)

**Keywords:** Bi_2_O_3_, PbO, sintering temperature, mechanical properties, lead-bismuth eutectic alloy

## Abstract

In this paper, the effects of Bi_2_O_3_ doping on the mechanical properties of PbO ceramic pellets were studied. Different ratios of Bi_2_O_3_/PbO (i.e., *x*Bi_2_O_3_-(1−*x*) PbO, where *x* is 0, 1, 3, 5, or 7 wt.%) were fabricated and sintered at 570, 620, and 670 °C. Mechanical properties including density, hardness, flexural strength, and sintering of PbO were studied for each of the aforementioned compositions. Phase composition, microstructure, and the worn surfaces of the composites were characterized by scanning electron microscopy and X-ray diffraction (XRD). The XRD analysis revealed that a solid solution formed in the composite ceramic. The best suited conditions of temperature and doping of Bi_2_O_3_ for optimal sintering were found to be 620 °C and 3 wt.%, respectively. The hardness of the 3 wt.% Bi_2_O_3_-97 wt.% PbO ceramic was found to be 717 MPa, which is about four times higher than the hardness of pure PbO. In addition, the strength of the composites was found to be 43 MPa, which is two times higher than that of pure PbO. The integrity of the composites was verified using the lead–bismuth eutectic alloy flushing experiment. The results of this research paper are important for future studies of oxygen control in the lead–bismuth eutectic alloy of lead-cooled fast reactors.

## 1. Introduction

Lead oxide (PbO) ceramic pellets are used in solid-phase oxygen control systems, which are an important constituent of lead-cooled fast reactors (LFRs). In LFRs, the lead–bismuth eutectic (LBE) alloy was chosen as a coolant material [1]. However, LBE can cause severe corrosion of structural materials [2], including pipes and other components. To prevent such corrosion, it is important to consider appropriate protective methods. Studies in the past have shown that oxygen is the most critical element in corrosion caused by LBE alloys. When oxide layers of a certain thickness get coated onto the internal surface of steel pipes, forming a film of magnetite ferroferric oxide and iron chrome spinel, $[Fe_{\left(3-x\right)}Cr_{x}]O_{4}$, they prevent the further penetration of LBEs. The thickness of the oxide layers vary at different levels of oxygen concentration; an oxide layer that is too thick or too thin cannot protect the steel well [3]. Some researchers have reported that solid-phase oxygen control is a promising anti-corrosion method [4]. Using this approach, PbO ceramic pellets are placed inside the pipes, and lead ions and oxygen ions are released from the ceramic pellets into the metal. The concentration of oxygen ions in the liquid metal can be adjusted by altering the flow rate and temperature of the coolant. To keep the coolant pure and to protect the steel pipe, high-strength PbO ceramic pellets are required, which have fast oxygen concentration regulation and crack resistance during operation.

Currently, experiments on solid-phase control systems are being carried out in the LBE test loop CRAFT at SCK·CEN (Studiecentrum voor Kernenergie/Centre d’Etude de l’Energie Nucléaire) in Mol, Belgium [5], in the Pb-Bi forced convection loop in Ibaraki, Japan [6], and as part of the DEMETRA project by CEA (Commissariat à l’énergie atomique et aux énergiesalternatives), Saclay in Gif-sur-Yvette, France [7]. Though several studies have been done theoretically and practically on solid-phase control systems, only a few have reported on the mechanical properties and microstructures of PbO [5,6,7,8]. Therefore, research on developing sintering techniques with PbO ceramic pellets that have excellent performance during operation has become a hot topic in this area.

The PbO ceramic pellets tested in the CRAFT loop developed cracks at their equator during the experiment (Figure 1) [5]. PbO ceramic pellets that break in the loop can result in debris being formed in the coolant, eventually blocking the pipeline in extreme cases [5]. Kondo et al. found that PbO powder was sintered in the form of lumps at 800 °C, following which the lumps were mechanically broken into small pieces [6]. The relative density of the PbO ceramic pellets in the above loop was 72.82%. Brissonneau reported that the PbO pellets were generated from PbO powder and water under a force of 45 kN for 2 min, and then the pellets were sintered at 620 °C for 2 h at a slow temperature ramp (2 °C·min^−1^) [7]. The ceramic pellets discussed above are used as PbO pellets in oxygen concentration control experiments, but there is no detailed literature on the properties of PbO ceramic pellets. It can be seen from Figure 1 that the PbO ceramic pellets used in the experiment were not high in strength and could be easily cracked along the equatorial plane because of their spherical shape.

We have increased the scope of our research to improve the mechanical properties of PbO ceramic pellets and prevent cracking in the LBE alloy. Doping by Pb powder using liquid-phase sintering improved the mechanical properties of the ceramic pellets [8]. The mechanical properties and microstructures of the Pb, PbO, and Pb_3_O_4_ ceramic pellets were also studied in detail, and the results showed that the addition of lead powder facilitates sintering and increases the strength of the ceramic. We also used the microwave sintering technique to improve the strength of the PbO ceramic pellets [9]. However, there was no significant improvement in the mechanical properties, and the technique resulted in the occurrence of cracks.

In this study, Bi_2_O_3_ powder as the sintering aid was added to the PbO precursor. Bi_2_O_3_ powder was chosen as the sintering agent for two reasons: (1) The raw material of the ceramic pellets will not pollute the coolant or make it impure and (2) the diameter of Bi^3+^ is close to the diameter of Pb^2+^, and their chemical bond properties are similar, which is beneficial in the sintering process. The addition of Bi^3+^ ions can selectively occupy Pb^2+^ positions by creating defects and vacancies. In this paper, the mechanical properties and microstructures of Bi_2_O_3_/PbO with different compositions (i.e., *x*Bi_2_O_3_-(1−*x*) PbO, where x is 0, 1, 3, 5 or 7 wt.%) were investigated in terms of their strength, hardness, and scanning electron microscopy (SEM) data to arrive at the optimal value of *x* and the ideal sintering temperature. Flushing experiments were then performed to verify the integrity of the PbO ceramic pellets.

## 2. Materials and Methods

The chemical composition of the ceramic was chosen to be *x*Bi_2_O_3_-(1−*x*)PbO (the value of *x* is 0, 1, 3, 5 or 7 wt.%) for conventional sintering. The initial Bi_2_O_3_ and PbO powders (A.R. 99% purity) were purchased from Sinopharm Chemical Reagent Co. Ltd., Beijing, China. The powders were mixed and ground using a planetary ball mill (FOCUCY, Changsha, China) with 2 wt.% deionized water for 24 h. The green compact was fabricated using a cold steel die at a uniaxial pressure of 40 MPa. The rate of pressure during compression and depression was kept below 0.5 MPa/s in order to reduce the elastic after effect and avoid splitting the compact. The specimens were then sintered at 570, 620 and 670 °C for 2 h at a heating rate of 10 °C·min^−1^, and were then cooled to room temperature [8].

During sintering, the compacts were covered by the PbO powder to further minimize PbO volatilization [10]. It was noticed that the loss of PbO lead to a decrease in density and affected the mechanical properties of the PbO ceramic pellets [11]. Disc samples (diameter, 13 mm) and bar samples (4 mm × 3 mm × 35 mm) were fabricated for mechanical property testing. The tablet-like pellets of PbO ceramic pellets (diameter, 6 mm; height, 5 mm) were fabricated to test their integrity using the flushing experiment.

Microstructures were observed using a Quanta 200F field emission scanning electron microscope (FEI, Zürich, Switzerland). The crystalline phases were analyzed using a D8 Focus X-ray diffraction (Bruker, Billerica, MA, USA) with Cu Kα radiation. Hardness was estimated on polished samples by the indentation method using a Micro Vickers Hardness Tester (Qualitest, Lauderdale, FL, USA). The indentation test was conducted with a load of 0.98 N held for 15 s. The hardness (Hv) was calculated according to ASTM C1327-15 [12] as below:(1)$$Hv=\frac{1.8544F}{d^{2}}$$
where *F* is the applied load and d is the mean value of the diagonal length of the indentation. A flexural strength test was conducted by the three-point flexural method (span, 35 mm; crosshead speed, 0.5 mm·min^−1^) using a WDW-100E computer controlled electronic universal testing machine (TIME Group Inc., Beijing, China). The densities of the specimens were estimated using the Archimedes method by weighing them in air and water. Data were collected for 6–8 samples to obtain an average of the results.

## 3. Results and Discussion

### 3.1. Mechanical Properties of Bi_2_O_3_-PbO Ceramic Pellets

Figure 2a shows the relative densities versus sintering temperatures for the PbO ceramic pellets as a function of Bi_2_O_3_ wt.%. The effect of sintering temperature on the density do not show a specific increasing or decreasing trend. The peak value of 92.01% is achieved at 670 °C with 1 wt.% Bi_2_O_3_. As a whole, excluding the effects of errors, the best density is achieved at a sintering temperature of 620 °C. From Figure 2a, we observe that with the increase in the sintering temperature and the Bi_2_O_3_ content the sintering process is prolonged, creating more pores and reducing the relative density of the ceramic pellets.

At each sintering temperature, the relative density of the ceramic pellets decreases with the increase in the Bi_2_O_3_ content. At a sintering temperature of 620 °C and an optimum doping value of 3 wt.% Bi_2_O_3_, a high relative density of 90.57% is achieved. Kondo and Takahashi [6] found that the relative density is 72.82% for pure PbO ceramic pellets sintered at 800 °C, which is lower than that found in this study. The results indicate that doping by a small amount of Bi_2_O_3_ significantly improves the density of PbO ceramic pellets.

Figure 2b shows a graph displaying flexural strength compared to Bi_2_O_3_ wt.% at different sintering temperatures for PbO ceramic pellets. The effect of the sintering temperature on the flexural strength is very prominent and changes significantly with the changing temperature. The flexural strength of the PbO ceramic pellets sintered at 620 °C is much higher than that of the samples sintered at 570 and 670 °C.

It is well known that pure PbO ceramic pellets are not tough enough to flush in LBE [5]. After doping with 1 wt.% Bi_2_O_3_ at 620 °C, the flexural strength of the PbO ceramic pellets increased by 75%. It reached a peak value of 43 MPa at 3 wt.% Bi_2_O_3_ and then tended to decrease slightly with further increases in the Bi_2_O_3_ content. At 570 °C, the strength of the ceramic pellets decreased abruptly when the Bi_2_O_3_ increased to 1 wt.%, but as the Bi_2_O_3_ content continued to increase, the strength also increased and attained a constant value of flexural strength. However, at 670 °C the opposite was true. The addition of 1 wt.% of Bi_2_O_3_ caused an increase in flexure strength, but further increases of the modifier resulted in the gradual lowering of flexure strength. The relationship between strength and porosity is defined by the Griffith equation [13] as shown below:(2)$$\sigma_{f}=\frac{{\left(1-\frac{P}{P_{0}}\right)}^{n}}{Y\sqrt{c}}$$
(3)$$c=c_{0}[1-\frac{\left(P_{0}-P\right)}{3(1-P)}]$$
where *P*_0_ and *n* are constants, *P* is the porosity, *Y* is the shape factor, and *c*_0_ is the initial crack length. Thus, it may be concluded that the strength of the sample is inversely proportional to its porosity and directly proportional to its density. 

The porosity becomes a dominant factor in decreasing strength when the Bi_2_O_3_ content is increased from 3 to 7 wt.% at 620 °C or at a higher sintering temperature of 670 °C, which is consistent with the results shown in Figure 2a.

The hardness of the PbO ceramic pellets at different wt.% of Bi_2_O_3_ as a function of temperature is plotted in Figure 3. It is obvious that the hardness of Bi_2_O_3_-PbO sintered at 620 °C is higher than that at 570 and 670 °C. At 620 °C, the change in hardness is very steep and increases gradually until the content of Bi_2_O_3_ is 3 wt.%. After reaching its peak, the hardness starts to decline. The optimum hardness of 717 MPa was achieved with 3 wt.% Bi_2_O_3_ at 620 °C, which is about four times higher than that of pure PbO.

### 3.2. Flushing Test

Suitable fabricated samples were obtained with a composition of 3 wt.% Bi_2_O_3_-97 wt.% PbO, sintered at 620 °C. They were then tested in an experimental apparatus using a solid-phase oxygen control system at the Beijing Key Laboratory of Passive Safety Technology for Nuclear Energy of North China Electric Power University (Figure 4).

In the flushing experiment, 135 ceramic samples of 3 wt.% Bi_2_O_3_-97 wt.% PbO were placed in the middle of the mass exchanger. LBE flowed through the PbO ceramic pellets at a speed of 110 L/h for 100 h. The flushing test was carried out at 450 °C, and the oxygen concentration was 3.14 × 10^−4^ wt.%.

Figure 5 shows the images of the PbO pellets before and after testing. None of the ceramic pellets cracked, which indicates that the strength of the 3 wt.% Bi_2_O_3_-97 wt.% PbO ceramic pellets sintered at 620 °C is promising in terms of meeting the needs of engineering strength requirements.

### 3.3. Microstructures of Bi_2_O_3_-PbO Ceramic Pellets

To understand the toughness mechanisms of doping Bi_2_O_3_, XRD was used to obtain the patterns of the PbO ceramic pellets sintered at 620 °C with different wt.% of Bi_2_O_3_ (Figure 6). The formation of massicot (PbO, PDF No. 72-0094) was identified as the main crystalline phase, with a small amount of litharge (PbO, PDF No. 85-0711). The litharge phase disappeared when the doping was done with more than 5 wt.% Bi_2_O_3_, and the intensity of massicot peaks became stronger. The crystals in the massicot phase were orthogonal (a = 5.489, b = 4.775, c = 5.891), and those in the litharge phase were tetragonal (a = 3.973, b = 3.973, c = 5.022). These are two forms of PbO that appear in nature. The phase transition temperature was 500 °C. The addition of aid promotes the phase transition process, so at high aid content, the main component is massicot. Lead bismuth oxide and Bi_2_O_3_ phases were not detected in the XRD patterns for the fabricated samples.

Figure 6 indicates that the peaks of the PbO phase shift to a large angle, which is attributed to the formation of a solid solution between Bi_2_O_3_ and PbO. The radius of Bi^3+^ (1.03 Å) is smaller than that of Pb^2+^ (1.19 Å). When Bi^3+^ ions are substituted with Pb^2+^ ions, there is a lattice distortion, and the resistance to dislocation increases. Therefore, the enhancement of mechanical properties is due to the strengthening of the solid solution of Bi^3+^ ions.

Figure 7a–j shows the SEM micrographs and fracture images of PbO ceramic pellets sintered at 620 °C with different wt.% of Bi_2_O_3_.

When doping is done with 1 wt.% Bi_2_O_3_, the crystals are observed to start growing (Figure 7d, indicated by frame). When the Bi_2_O_3_ content increases to 3 wt.%, grains are seen to aggregate to form distinct grain boundaries (Figure 7e). When the Bi_2_O_3_ content continues to increase, a solid solution is formed, wherein the solute ions Bi^3+^ replace the host ions Pb^2+^, resulting in the formation of positive ion vacancies, which is beneficial for the sintering process and for metallurgical binding [14,15].

Based on the results shown in Figure 2, *x*Bi_2_O_3_-(1−*x*) PbO (*x* being 0–3 wt.%) can be considered to have the same relative density of ~90.6%. Thus, the doping of Bi_2_O_3_ intensifies the sintering process by grain growth at 620 °C as the amount of Bi_2_O_3_ is increased from 0 to 3 wt.%. According to Equations (2) and (3), it is seen that the effect of porosity on the performance of ceramic pellets is dominant. It can be seen from Figure 7i that pores are wrapped in grains, indicating that the grain boundaries move rapidly with 7 wt.% Bi_2_O_3_ at 620 °C, which leads to a decline in its mechanical properties. In addition, it can be seen from Figure 2 that the density of Bi_2_O_3_-PbO ceramic pellets drops sharply when the Bi_2_O_3_ content exceeds 3 wt.%.

Figure 8 presents the SEM micrographs and fracture images of 3 wt.% Bi_2_O_3_-97 wt.% PbO sintered at 570 and 670 °C. In addition, obvious powder agglomeration is observed in Figure 8a,b, indicating that sintering is not complete at this low temperature (570 °C). Thus, grain growth plays a leading role in enhancing hardness and flexural strength at lower temperature (570 °C). Thus, the optimum sintering temperature and doping amount of Bi_2_O_3_ for Bi_2_O_3_-PbO ceramic pellets are determined to be 620 °C and 3 wt.%, respectively.

## 4. Conclusions

In this paper, the effects of doping Bi_2_O_3_ and differing sintering temperatures on the mechanical properties of PbO ceramic pellets were investigated. It was found that doping with a moderate amount of Bi_2_O_3_ can significantly enhance the mechanical properties of PbO ceramic pellets. The enhancement is attributed to the formation of a solid solution of Bi^3+^ ions. The optimal sintering temperature and doping amount of Bi_2_O_3_ are 620 °C and 3 wt.%, respectively, for PbO. The hardness increased four times and the strength increased two times compared with pure PbO. The ceramic pellets showed good performance in the LBE flushing experiment.

## 5. Patents

The method of bismuth oxide reinforced lead oxide ceramics introduced in this paper has been applied to China Patent Office for an invention patent.

## Figures and Tables

**Figure 1 materials-12-01948-f001:**
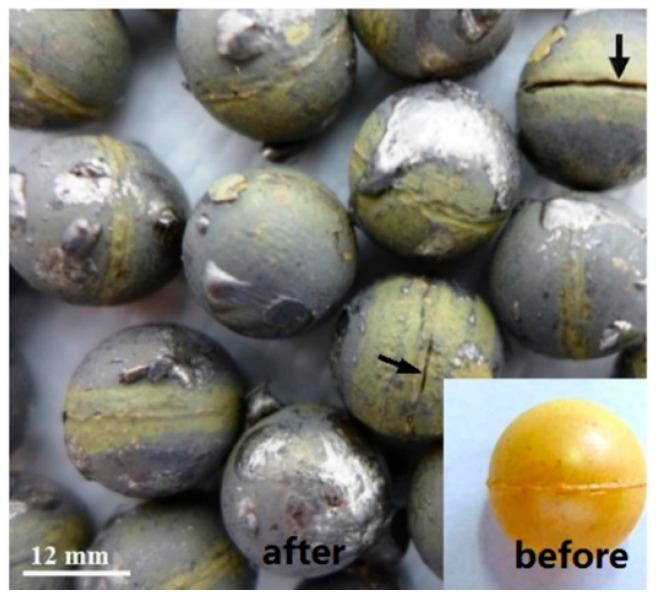
PbO ceramic pellets tested in the CRAFT loop and MEXICO loop [5].

**Figure 2 materials-12-01948-f002:**
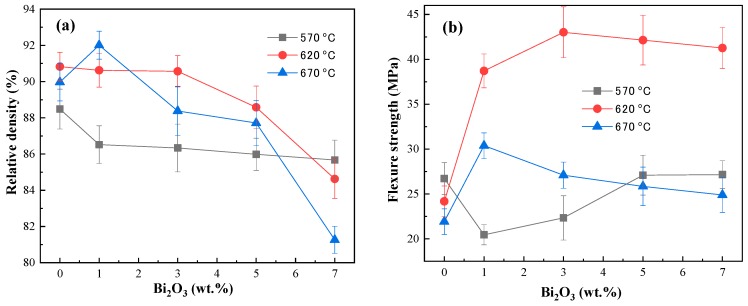
(**a**) Relative density and (**b**) flexure strength versus sintering temperature for PbO ceramics with different Bi_2_O_3_ wt.%.

**Figure 3 materials-12-01948-f003:**
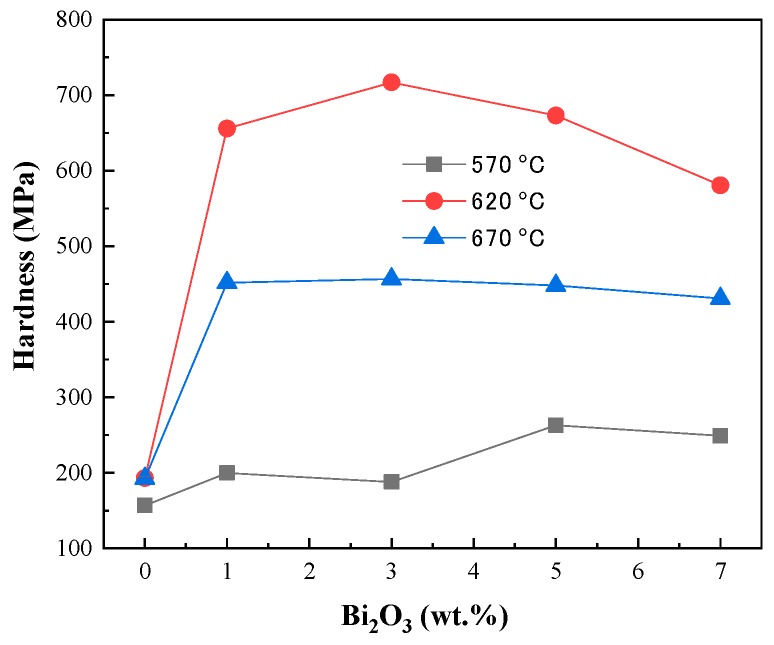
Hardness versus sintering temperature of PbO ceramic pellets with different wt.% of Bi_2_O_3_.

**Figure 4 materials-12-01948-f004:**
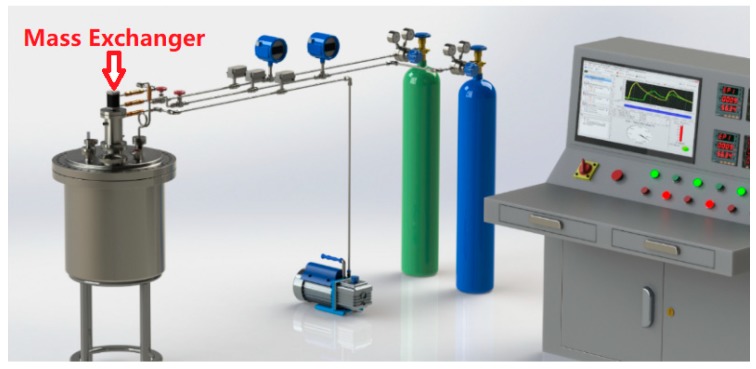
The experiment bench of the oxygen control system.

**Figure 5 materials-12-01948-f005:**
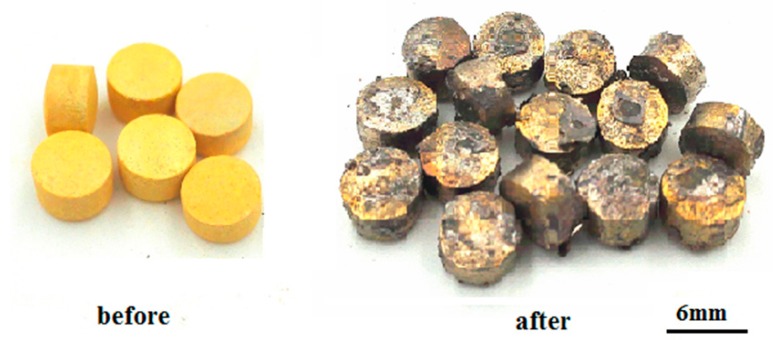
The 3 wt.% Bi_2_O_3_-97 wt.% PbO pellets before and after the flushing experiment.

**Figure 6 materials-12-01948-f006:**
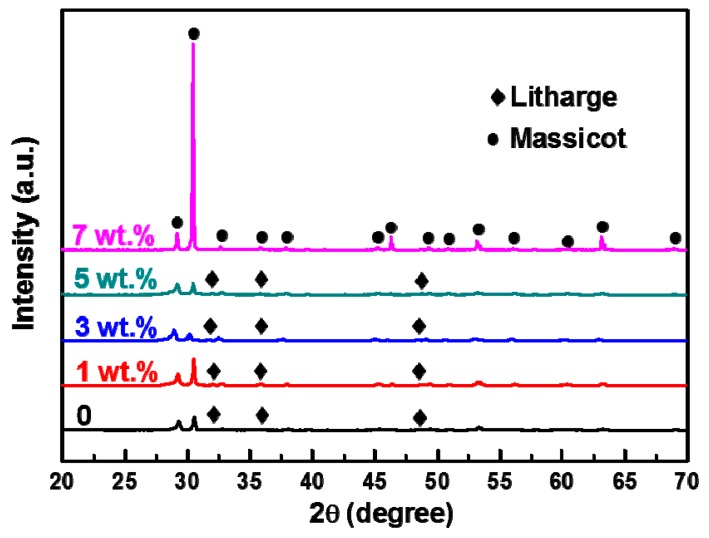
XRD patterns of PbO ceramic pellets sintered at 620°C with different wt.% of Bi_2_O_3_.

**Figure 7 materials-12-01948-f007:**
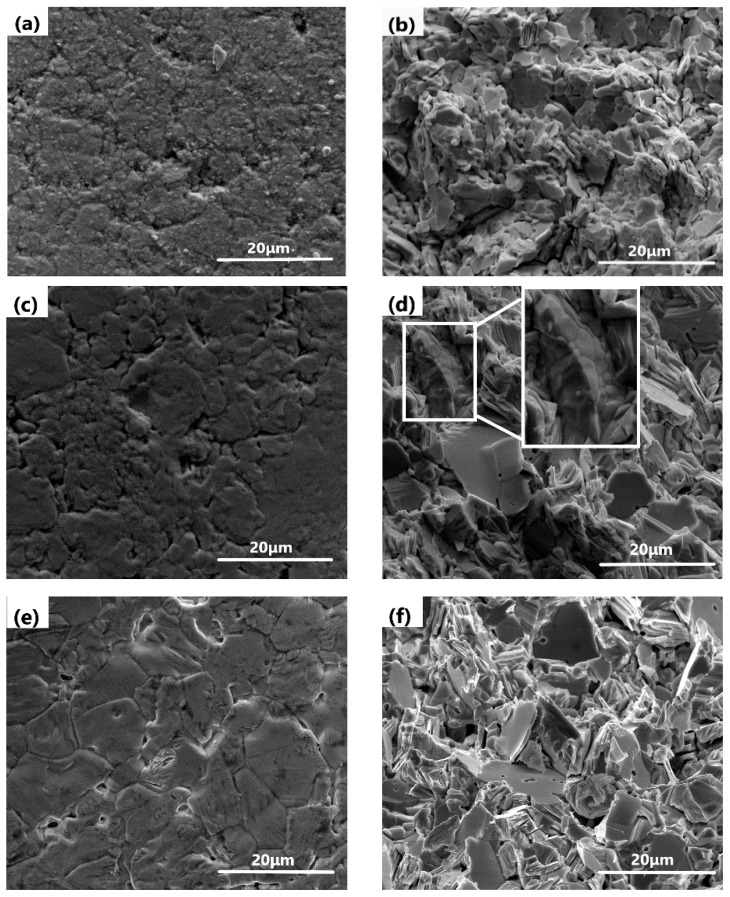
SEM micrograph and fracture images of PbO ceramic pellets sintered at 620 °C with different Bi_2_O_3_ content: (**a**,**b**) 0 wt.%, (**c**,**d**) 1 wt.%, (**e**,**f**) 3 wt.%, (**g**,**h**) 5 wt.%, and (**i**,**j**) 7 wt.%.

**Figure 8 materials-12-01948-f008:**
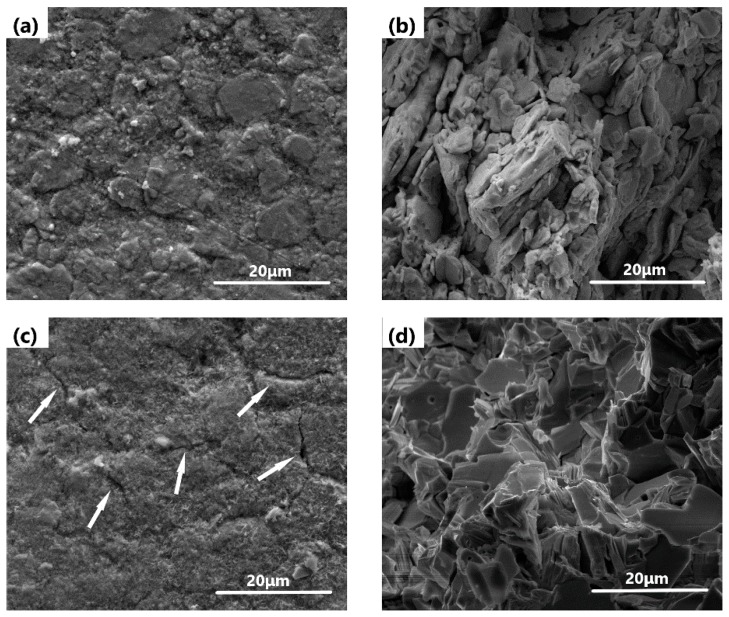
SEM micrographs and fracture images of 3 wt.% Bi_2_O_3_-97 wt.% PbO sintered at (**a**,**b**) 570 °C, and (**c**,**d**) 670 °C.

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
