# Peer review of "Effects of Bi2O3 Doping on the Mechanical Properties of PbO Ceramic Pellets Used in Lead-Cooled Fast Reactors"

_materials, 2019, doi:10.3390/ma12121948_

Reviewer 1 Report

-

Author Response

We sincerely thank the reviewer for the valuable feedback that we have used to improve the quality of our manuscript. We here provide a point-by-point response to the reviewer’s comments. 

 Point 1: “In this study, Bi2O3 powder as the sintering aid was added to the PbO precursor. Bi2O3 powder was chosen as the sintering agent for two reasons: (1) the raw material of the ceramic pellets will not cause pollute the coolant or make it impure and (2) the diameter of Bi3+ is similar to the diameter of Pb2+  and their chemical properties are very similar, which is beneficial in the sintering process.” I can not agree with information about the strong similarities between chemical properties of Pb2+ and Bi3+. The elements belong to the two different groups and have different valency. The addition of bismuth changes the space charge distribution in the sample and affects the change of its electrical properties by creating defects and vacancies (Adamczyk, M., Ujma, Z., Szymczak, L., Koperski, J “Influence of post-sintering annealing on relaxor behaviour of (Pb 0.75 Ba0.25 )(Zr0.70 Ti0.30 )O3 ceramics”, 2005 Ceramics International 31(6), pp. 791-794)  

 Response 1: I revise this paragraph (Line72-76):  In this study, Bi2O3 powder as the sintering aid was added to the PbO precursor. Bi2O3 powder was chosen as the sintering agent for two reasons: (1) the raw material of the ceramic pellets will not cause pollute the coolant or make it impure and (2) the diameter of Bi3+ is close to the diameter of Pb2+ and their chemical bond properties are similar, which is beneficial in the sintering process. The addition of Bi3+ ions can selectively occupy Pb2+ positions, by creating defects and vacancies. 

 Point 2: The Authors presented the part results of the research in a clear and understandable way. Interpretation of the obtained results is correct and interesting, whereas its value would increase by references to the results of investigations carried out by other scientists. The Authors have not avoided a small inaccuracy, writing: “It increased linearly at 1 wt% Bi2O3 and then started to decrease and continued to decrease.” 

 Response 2: I change the sentence (Line138-139) as reviewer’s kind suggestion: “Addition of 1 wt% of Bi2O3 caused increase of flexure strength. Further increase of the modifier results in gradual lowering of the discussed value.”

 Point 3: The reviewer suggest measuring the porosity using the Archimedes method. 

 Response 3: Porosity%=100%-relative desity% Values of relative density have been shown in Fig.2 (a).

 Point 4: I can not understand why the second part of the results the Authors posted in the chapter ”Discussion”. In my opinion this is illogical. I suggest combining two types (“Results” and“Discussion”) into one under a common name “Results and discussion”.

 Response 4:Follow your suggestion, I combine two types (“Results” and“Discussion”) into one under a common name “Results and discussion”.

Reviewer 2 Report

It would be important to determine the particle size distribution of the initial oxide powders.

Have you even considered the option of adding small quantity of additives?

Author Response

We sincerely thank the reviewer for the valuable feedback that we have tried to test the properties by adding small quantity of additives.

 Thank you very much.

Reviewer 3 Report

The authors present a microstructural and mechanical characterization of PbO +0-7% Bi2O3 samples sintered at different temperatures. Although the motivations are of interest the experiments have serious flaws listed in the following:

1) The used Archimede method is not realiable for the density measurement. In fact, samples with highy porosity and surface breaking pores or cracks, such that shown in fig 8, allow for water-infiltration may result in erroneous measurements.

2) The microstructure analysis should be done on polished surfaces rather than on as-sintered and fractured surfaces showed in fig 7 and 8. Moreover since there is some speculation on the grain size, this parameter should be measured. It could be useful shown also some micrograph of the indented surfaces.

3) All the measures should have the errors.

4) Lack of references. Sentence as: "Many reserach studies...". "only a few have reported on the mechanical properties...", "Some studies..." should be reported the proper references.

5) As done for density, the literature's strength value of PbO should be reported.

Author Response

We sincerely thank the reviewer for the valuable feedback that we have used to improve the quality of our manuscript. We here provide a point-by-point response to the reviewer’s comments.

Point 1: The used Archimede method is not realiable for the density measurement. In fact, samples with highy porosity and surface breaking pores or cracks, such that shown in fig 8, allow for water-infiltration may result in erroneous measurements.

Response 1: We agree that Archimede method is not realiable for the density measurement. At present, we do not have the necessary tool-set to study the accurate density of PbO ceramics. This method was used for obtain a compared results of density by reference to a number of literatures. [1] Adamczyk, M., Ujma, Z., Szymczak, L., Koperski, J., Influence of post-sintering annealing on relaxor behaviour of (Pb0.75 Ba0.25 )(Zr0.70 Ti0.30 )O3 ceramics, Ceramics International, 2005, 31(6), pp. 791-794;  [2] ADAMCZYK M., UJMA Z., SZYMCZAK L.. et al. Effect of Nb doping on the relaxor behaviour of (Pb0.75Ba0.25)(Zr0.70Ti0.30)03ceramics.J Eur Ceram Soc,2006,26(3), pp. 331-336; [3] Odile Ast, Marc Perez, Sebastien Carlet, PuAl alloys density measurements using gas pycnometer: First results, J. Alloys & Compounds, 2007, 444–445( 11 ), pp. 226-229; [4] Adamczyk M., Molak A., Ujma Z. The influence of axial pressure on relaxor properties of BaBi2Nb2O9 ceramics, Ceramics International, 2009, 35, pp. 2197–2202.

Point 2: The microstructure analysis should be done on polished surfaces rather than on as-sintered and fractured surfaces showed in fig 7 and 8. Moreover since there is some speculation on the grain size, this parameter should be measured. It could be useful shown also some micrograph of the indented surfaces.

Response 2: We have tried many times to treat the as-sintered sample by polishing, hot corrosion or acid corrosion in the surface observation, but the SEM images are not good. So the pictures selected in the paper are unpolished and acid. The grain size of pure PbO is about 8.2-8.6 microns, and the grain size after the aid content exceeds 3% is basically unchanged.

Point 3: All the measures should have the errors.

Response 3: We add the errors in the corresponding Fig2 (a) and (b). It is too low to indicate in Fig.3.

Point 4: Lack of references. Sentence as: "Many reserach studies...". "only a few have reported on the mechanical properties...", "Some studies..." should be reported the proper references.

Response 4: Some expressions, such as many \a few\ some, are not proper. Actually, the references are not many. As suggested by the reviewer, we add references or edit our words again.

Line 39: Some researchers reported that solid-phase oxygen control is a promising anti-corrosion method [4].

Line 49: Though several studies have been done theoretically and practically on solid-phase control systems, only a few have reported on the mechanical properties and microstructures of PbO [5-6][8-9].

Line 53: The study has reported that the PbO ceramic pellets tested in CRAFT loop were cracked in the area of equator during the experiment (Fig. 1) [5].

Point 5: As done for density, the literature's strength value of PbO should be reported.

Response 5: Thanks for your suggestion.

 At present, there are few studies on lead oxide ceramics in solid-state oxygen control systems. The density of lead oxide is only mentioned in reference [6], i.e. 6.94g/m3, which convert into relative density 72.82% for purpose of comparison. We refered it in the result section (see line 126).

The value of flexual strength test by the three-point flexural method is not mentioned in the references.

Round  2

Reviewer 1 Report

The authors corrected manuscript accordingly with the suggestions of reviewer, took into account all comments and hints. In my opinion the article includes innovative results and could be of interest to the readers and now is free from minor methodological errors and text editing. Therefore, I recommend manuscript for publication in present form.

Author Response

Thank you.

Reviewer 3 Report

The authors did not respond to all the raised point, in particular:

Point 1) I suggest to measure the relative density by boiling water method and/or image analysis of polished sections. The authors may try also the dimensional method. The density values are one of the main results of this article and should be reliable.

Point 2) All the presented conclusion should be supported by experimental data. The authors did not measure the grain size distribution. Sentence about the Hall–Petch equation, as well as the following sentences, should be removed:

- “First, with the addition of Bi2O3, the grain size of PbO ceramic pellets increases as compared with the pure PbO ceramic pellets. The grain boundaries cannot be seen clearly, which indicates that the sample is at the initial stage of sintering and metallurgical binding has not yet occurred between the powders”.

-“ There is almost no change in the size of the grains for 3–7 wt% Bi2O3 ceramic pellets as observed by SEM (Fig. 7(e–j)).”

-“ In addition, it can be seen from Fig. 2 that the density of Bi2O3–PbO ceramic pellets drops 210 sharply when the Bi2O3 content exceeds 3 wt%, with almost the same grain size.”

Also the sentence:  “However, at 670 °C, microcracks are observed on the surface of 3 wt% Bi2O3–97 wt% PbO ceramic pellets (Fig. 8c). They are apt to appear at the interface of two ceramic grains and readily propagate through the grains, which causes their hardness and strength to be lower than those of the ceramic pellets at 620 °C.” should be deleted since all the presented as-sintered surfaces show “microcracks” between adjacent grains.

Author Response

We sincerely thank the reviewer for the valuable feedback that we have used to improve the quality of our manuscript. We here provide a point-by-point response to the reviewer’s comments.

Point 1: I suggest to measure the relative density by boiling water method and/or image analysis of polished sections. The authors may try also the dimensional method. The density values are one of the main results of this article and should be reliable.

Response 1: Thanks very much for your suggestion. Actually, the Archimede method we used for the density measurement is the same process as by boiling water method.

Our measurement process of relative density is as follows.

1.      Dry the samples to constant weight and heating to 110 for at least 4 h. Determine the dry weight, D, in grams to the nearest 0.01g.

2.      Place the samples in water and boil for 2 h. Then soak for 12h.

3.      Determine the suspended weight, S, of each sample after boiling and while suspend in water, in grams to the nearest 0.01g. Keep the sample entirely covered with water, and allow no contact with the bottom and wall of the container. All samples are immersed at the same depth.

4.      Saturated weight, W – after determining the suspended weight, blot each sample with a moistened cotton cloth to remove drop ship of water from the surface and determine the saturated weight, W, in grams to the nearest 0.01g.

5.      Calculate bulk density ρW:

6.      Calculate theoretical density ρ0:       

X and Y are the mass fractions of the constituents of the sample, respectively, and ρ1 and ρ2 are the theoretical densities of the corresponding components.

7.      Calculate relative density R:               

Point 2: All the presented conclusion should be supported by experimental data. The authors did not measure the grain size distribution. Sentence about the Hall–Petch equation, as well as the following sentences, should be removed:

- “First, with the addition of Bi2O3, the grain size of PbO ceramic pellets increases as compared with the pure PbO ceramic pellets. The grain boundaries cannot be seen clearly, which indicates that the sample is at the initial stage of sintering and metallurgical binding has not yet occurred between the powders”.

-“ There is almost no change in the size of the grains for 3–7 wt% Bi2O3 ceramic pellets as observed by SEM (Fig. 7(e–j)).”

-“ In addition, it can be seen from Fig. 2 that the density of Bi2O3–PbO ceramic pellets drops 210 sharply when the Bi2O3 content exceeds 3 wt%, with almost the same grain size.”

Also the sentence:  “However, at 670 °C, microcracks are observed on the surface of 3 wt% Bi2O3–97 wt% PbO ceramic pellets (Fig. 8c). They are apt to appear at the interface of two ceramic grains and readily propagate through the grains, which causes their hardness and strength to be lower than those of the ceramic pellets at 620 °C.” should be deleted since all the presented as-sintered surfaces show “microcracks” between adjacent grains.

Response 2:  We agree with your suggestion. We removed the sentences, see lines: 145-147, 193-196, 205-206, 211, 215-219. Also we delete Ref [14] (the Hall–Petch equation).